# OpenReview forum: "Why does Weak-OOD Help? A Further Step Towards Understanding Jailbreaking VLMs"
_ICLR.cc/2026/Conference — ICLR 2026 Conference Withdrawn Submission_

### Official Review · Reviewer_Nqfw · 2025-10-30

**Soundness:** 2
**Presentation:** 3
**Contribution:** 2
**Rating:** 2
**Confidence:** 5

**Summary:**

This paper investigates Out-of-Distribution (OOD)-based jailbreak attacks on large Vision-Language Models (VLMs), focusing on a phenomenon called weak-OOD. The authors reaffirm that mild OOD perturbations to images (e.g., shuffling or mixup) often improve jailbreak success rates, whereas stronger perturbations degrade attack performance. The paper attributes this weak-OOD effect to a trade-off between two internal mechanisms within safety-aligned VLMs: mild OOD perturbations maintain malicious intent perception (via staying close to the pre-training distribution) while weakening the refusal trigger (via naturally lying far from the narrow alignment training distribution). Building on this understanding, the paper introduces a new attack that exploits discrepancy of OCR capability induced from the mismatched generalization between pre-training and safety-alignment.

**Strengths:**

- The paper reaffirms the vulnerability of VLMs with regard to weak OOD harmful inputs and the mismatched generalization between pre-training and safety-alignment training.
- The proposed method extends prior work (i.e., FigStep) which leverages the poor generalization of safety-alignment on OCR harmful inputs, showing great jailbreak effectiveness even with simple modifications (e.g., font size variation) from the baseline (FigStep).
- The paper proposed a mechanistic quantitative framework to measure how OOD manipulations affect intent preservation and refusal triggering in VLMs.
- The paper is well-written and easy to follow.

**Weaknesses:**

- The paper’s central hypothesis that the weak-OOD phenomenon arises from the asymmetry between pre-training and alignment is not conceptually new. Similar insights have already been discussed in the previous work (https://arxiv.org/abs/2307.02483), which tackles mismatched generalization that arises when inputs are out-of-distribution for a model’s safety training data but within the scope of its
broad pretraining corpus.
- This paper largely reiterates previously reported empirical findings rather than uncovering a new behavioral pattern, offering limited novelty in its characterization of the phenomenon: the weak-OOD phenomenon itself — where mild OOD perturbations preserve intent perception while lowering refusal rates, but stronger OOD disrupts intent understanding — was already empirically observed in prior work, such as JOOD.
- Moreover, while the paper stresses the importance of weak-OOD manipulation, JOCR provides no principled procedure for generating or identifying weak-OOD inputs, relying instead on marginal modifications (e.g., font and spacing variations) on the baseline (FigStep). Such ad-hoc tweaks still might not offer guarantee of intent preservation, possibly yielding strong-OOD inputs.
- (Minor) Some typos: "the its" in L81, "is more than the distance" in L385

**Questions:**

See above weaknesses.

---

### Official Review · Reviewer_tECe · 2025-10-30

**Soundness:** 2
**Presentation:** 3
**Contribution:** 2
**Rating:** 4
**Confidence:** 4

**Summary:**

This paper investigates why “weak” Out-of-Distribution (OOD) perturbations are surprisingly effective in jailbreaking Vision-Language Models. The authors design JOCR, a new OOD-based jailbreak method that leverages the model’s OCR capability from pre-training. JOCR achieves state-of-the-art jailbreak success on multiple benchmarks (e.g., HarmBench, RedTeam-2K).

**Strengths:**

The paper provides a novel theoretical insight by identifying the pretrain–alignment inconsistency as the key reason why weak OOD perturbations can effectively bypass safety mechanisms.
 It demonstrates strong empirical rigor, with comprehensive ablations and benchmarks confirming that mild OOD manipulations outperform existing jailbreak methods.
The proposed JOCR attack introduces an innovative OCR-aware mechanism that not only validates the theory but also exposes practical vulnerabilities in real-world VLM safety alignment.

**Weaknesses:**

1. While the paper attributes the improved jailbreak success under mild perturbations to a weak-OOD effect arising from pretrain–alignment mismatch, an alternative explanation could be that the model is simply non-robust to low-level perturbations.
The experiments demonstrate that moderate perturbations yield higher attack success rates, but it is unclear whether this behavior necessarily reflects a semantic distribution shift rather than standard sensitivity to visual noise or encoding artifacts.

2. The proposed “weak-OOD” notion is very close to established ideas in robustness research, yet the paper does not clearly explain what is new beyond existing robustness frameworks.

3. If i do some perturbation and trained for robustness improvement, will that solve the issue?

4. It is not very clear of why this happens in practical. For evaluations, could you add some reasoning/interpretation so that it can understand more context?

**Questions:**

see weaknesses

---

### Official Review · Reviewer_jc87 · 2025-10-31

**Soundness:** 2
**Presentation:** 2
**Contribution:** 2
**Rating:** 4
**Confidence:** 3

**Summary:**

This paper investigates why weak-OOD manipulations (mild distributional shifts in jailbreak inputs) can outperform stronger perturbations when attacking VLMs. The authors identify a trade-off between input-intent perception and model-refusal triggering.

**Strengths:**

1. Paper is easy to follow and read.
2. The intuition of seeking weak-OOD in perturbation is straightforward.

**Weaknesses:**

1. Layer heterogeneity is ignored in the perception measurement. Simply doing the aggregation by averaging the Layer-Wise Cosine Similarity and Layer-Wise Refusal Similarity seems a bit too brutal.
2. Especially for the jailbreak part, the different settings in creating the perturbations are not provided with ablation studies (those from appx. tab. C). The impact of the settings on the degree of intent perception and on the refusal triggering is not quantified or further analyzed. I think this is an important part that is missing from the paper.

**Questions:**

See weakness.

---

### Official Review · Reviewer_Zt8k · 2025-10-31

**Soundness:** 2
**Presentation:** 3
**Contribution:** 2
**Rating:** 4
**Confidence:** 4

**Summary:**

This paper investigates why mild out-of-distribution perturbations (“weak-OOD”) often lead to higher jailbreak success rates on vision-language models (VLMs) than either no OOD or strong OOD perturbations. The authors analyze this effect through experiments on representative OOD jailbreak methods and show that moderate shifts preserve malicious-intent perception while weakening refusal behavior. They argue this arises due to misaligned robustness between pre-training (which enables stable intent perception) and safety alignment (more brittle refusal triggers). They then introduce JOCR, a jailbreak method that perturbs embedded text in ways consistent with OCR features learned during pre-training. JOCR achieves higher attack success rates than prior methods across multiple VLMs.

**Strengths:**

The paper focuses on a meaningful question—why weak OOD helps jailbreak VLMs, rather than only proposing another attack. The authors perform ablations across multiple attack methods and target models, illustrating the weak-OOD pattern with concrete quantitative trends and internal activation analyses. The mechanistic view separating intent perception and refusal triggers is supported by empirical signals across layers, and the theory connects naturally to pre-training versus alignment data distribution. Based on this insight, the paper builds JOCR, which demonstrates notable improvements over strong baselines. The argument is cohesive from empirical observation to conceptual explanation to method design.

**Weaknesses:**

Although the central idea is interesting, the theoretical component still feels heuristic. The latent-space interpretation depends on assumptions about token roles and feature locality that recent work debates; the evidence is more correlational than causal.

Some experimental choices raise questions about generalization. For example, the heavy reliance on shuffle-based perturbations when defining OOD magnitude, and the use of GPT-4o as a judge in evaluation, which may introduce bias or shortcuts.

The JOCR method is effective, but largely builds on a known idea (text embedding via typography) plus stochastic perturbations, so the novelty lies mainly in conceptual framing rather than methodology.

The work does not deeply explore defenses, which weakens practical impact for safety research.

**Questions:**

How stable is the weak-OOD effect across non-shuffle OOD directions, such as color jitter, geometric warps, or structured occlusion?

Does JOCR degrade substantially when OCR content is noisy or stylized beyond typical pre-training patterns?

---

### Note · Authors · 2025-12-24

I have read and agree with the venue's withdrawal policy on behalf of myself and my co-authors.